# Strong Plasmon–Exciton Coupling in Ag Nanoparticle—Conjugated Polymer Core-Shell Hybrid Nanostructures

**DOI:** 10.3390/polym12092141

**Published:** 2020-09-19

**Authors:** Christopher E. Petoukhoff, Keshav M. Dani, Deirdre M. O’Carroll

**Affiliations:** 1Femtosecond Spectroscopy Unit, Okinawa Institute of Science and Technology Graduate University, Onna, Okinawa 904-0495, Japan; kmdani@oist.jp; 2Department of Materials Science and Engineering, Rutgers University, 607 Taylor Rd., Piscataway, NJ 08854, USA; 3Department of Chemistry and Chemical Biology, Rutgers University, 610 Taylor Rd., Piscataway, NJ 08854, USA

**Keywords:** conjugated polymer, exciton, plasmon, vibrationally-dressed, strong coupling, electromagnetic simulations

## Abstract

Strong plasmon–exciton coupling between tightly-bound excitons in organic molecular semiconductors and surface plasmons in metal nanostructures has been studied extensively for a number of technical applications, including low-threshold lasing and room-temperature Bose-Einstein condensates. Typically, excitons with narrow resonances, such as *J*-aggregates, are employed to achieve strong plasmon–exciton coupling. However, *J*-aggregates have limited applications for optoelectronic devices compared with organic conjugated polymers. Here, using numerical and analytical calculations, we demonstrate that strong plasmon–exciton coupling can be achieved for Ag-conjugated polymer core-shell nanostructures, despite the broad spectral linewidth of conjugated polymers. We show that strong plasmon–exciton coupling can be achieved through the use of thick shells, large oscillator strengths, and multiple vibronic resonances characteristic of typical conjugated polymers, and that Rabi splitting energies of over 1000 meV can be obtained using realistic material dispersive relative permittivity parameters. The results presented herein give insight into the mechanisms of plasmon–exciton coupling when broadband excitonic materials featuring strong vibrational–electronic coupling are employed and are relevant to organic optoelectronic devices and hybrid metal–organic photonic nanostructures.

## 1. Introduction

Strong light-matter interactions involving organic semiconductors are important for a number of technical applications, including low-threshold lasing [1,2], room-temperature exciton-polariton Bose-Einstein condensates [3], and strongly-coupled organic microcavities [4,5]. Hybridization between excitons in organic semiconductors and photons in cavities results in the formation of polaritons, which have been observed experimentally and theoretically as energetic splitting in the normal modes of the coupled system, (i.e., Rabi splitting) [4,5]. Room-temperature, Rabi splitting values as large as 100 meV to 200 meV have been reported for organic microcavities [4,6,7], which are substantially larger than those observed for inorganic semiconductors within microcavities (~10 meV) [4,8] due to the larger binding energy of the Frenkel-type excitons in organic semiconductors [9].

Excitons in organic materials have also been shown to hybridize with surface plasmons, forming plasmon–exciton hybrid modes often called “plexcitons” [10,11,12,13]. Strong coupling between plasmons and excitons has had applications in optoelectronics [14,15], optical sensing [16], reversible switching [17], and control of chemical reactions [18,19]. For both photon–exciton and plasmon–exciton coupling, the coupling strength is given by the rate of energy exchange between the two oscillators, $g=\hbar \Omega /2=\int {\vec{\mu }}_{\mathrm{ex}}\cdot {\vec{E}}_{c}dV$, where *ħ*Ω is the Rabi splitting energy determined from the energetic splitting between the hybridized peaks in the absorption or scattering spectra of the coupled system, ${\vec{\mu }}_{\mathrm{ex}}$ is the transition dipole moment of the exciton and ${\vec{E}}_{c}$ is the cavity-confined electric field [8,20,21]. Plasmon–exciton coupling can thus be much stronger than photon–exciton coupling due to the extreme confinement of light to the nanoscale by coupling to surface plasmons [22], resulting in stronger electric field enhancements than what can be achieved using microcavities [13,23,24,25]. To achieve strong plasmon–exciton coupling, the splitting energy, *ħ*Ω, of the plasmon–exciton hybrid modes should be greater than the linewidths of the uncoupled plasmon and exciton resonances (i.e., *γ*_sp_ and *γ*_ex_, respectively) [21,26,27]. This suggests that extremely narrow plasmon and exciton resonances are necessary to achieve strong coupling. Most early studies have investigated the coupling between *J*-aggregates of organic dyes and localized surface plasmon resonances (LSPRs) using a core-shell geometry due to the narrow linewidths of *J*-aggregates (~50 meV) [28,29,30,31]. However, for device applications, such as plasmon-enhanced photovoltaics, light-emitting diodes, and spasers (the surface plasmon analogue to a laser), coupling between surface plasmons and excitons within conjugated polymers is of great interest.

Although conjugated polymers have broader spectral linewidths (50 meV to greater than 1 eV) [32,33,34,35] than typical excitonic materials employed in strong coupling studies, their large chromophore density manifests in very large ${\vec{\mu }}_{\mathrm{ex}}$ [1,36,37], which, when coupled to the strong electromagnetic fields from LSPRs, can potentially result in *ħ*Ω exceeding both *γ*_sp_ and *γ*_ex_. Much of the broadening observed in conjugated polymers arises from their strong electron-phonon coupling, whereby the electronic response of conjugated polymers is heavily influenced by vibrational modes [38,39]. There have been a few recent theoretical studies demonstrating that vibrationally-dressed excitonic states can strongly couple with microcavities [40,41,42]. We have previously demonstrated coupling between excitons in conjugated polymers and surface plasmons arising from metasurfaces [43,44], nanoparticle-on-mirror [45], and semiconductor-metal-insulator waveguide [46] geometries. Additionally, it has been demonstrated that when plasmonic nanostructures are embedded into broadband, conjugated polymer-fullerene photovoltaic absorber layers that the largest absorption enhancements occur at wavelengths slightly red-shifted from the absorption edge of the absorber (i.e., at the “red-edge”) [43,47,48,49,50,51,52]. This red-edge absorption enhancement occurs frequently for a wide range of different plasmonic-enhanced conjugated polymer photovoltaics, which suggests the pervasive presence of plasmon–exciton coupling, where the long wavelength plasmon–exciton hybrid mode leads to the strongest absorption enhancement. Here, we demonstrate, using electromagnetic simulations, that, despite the large absorption bandwidth and strong vibronic nature of conjugated polymers, plasmon–exciton coupling using realistically modeled conjugated polymers can result in ultra-strong coupling, with Rabi splitting energies reaching values greater than 1000 meV. We investigate plasmon–exciton coupling using a conventional core-shell hybrid nanostructure (Figure 1a), with spherical Ag nanoparticles (AgNPs) as the core of the hybrid structures and model conjugated polymers as the shell.

## 2. Materials and Methods

3D finite-difference time-domain (FDTD) simulations of the scattering and absorption from core-shell NPs were conducted using commercially available software (FDTD Solutions, Lumerical Solutions, Inc., Vancouver, BC, Canada) [53]. The AgNP core had a fixed radius of 25 nm, and its relative permittivity was obtained from the Handbook of Optical Constants (Palik), as provided by the software [53]. The relative permittivity for the conjugated polymer shells were input as analytical equations into the software using the equations described in Section 3.1 and the parameters described in the introduction. The excitation source used was a linearly polarized total-field scattered-field (TFSF) plane wave normally incident onto the core-shell NP. The TFSF source dimensions were fixed as 100 nm larger than the diameter of the core-shell hybrid NP. The wavelength range of the source was set to 300 nm to 900 nm with 5 nm increments, which corresponds to a source pulse length of 1.995 fs as a numerical parameter for the pulse. Continuous wave (CW) normalization was done after the completion of the time-domain simulations to obtain the steady-state response of the system. Perfectly matched layer (PML) boundary conditions were used in all directions, and the FDTD simulation region was fixed as 1000 nm larger than the diameter of the core-shell hybrid NP. The background permittivity of the simulations was set to that of water (1.77), unless otherwise specified. A mesh override region of 1.0 nm in all directions was used for the region surrounding the core-shell NP (note that for shell thicknesses greater than 40 nm, a 2.0 nm mesh override region was used over the entire core-shell structure, with a 1.0 nm mesh override region over the core and extending 10 nm in each direction into the shell). The simulations continued until either the fields decayed to 10^−5^ of their initial value, or after 100 fs of simulation time (typically the former).

For the simulations of actual conjugated polymer materials, the dispersive relative permittivity for each material was obtained from the following sources: Tamer et al. for MEH-PPV [54]; Morfa et al. for P3HT [55]; Bernhauser for PTB7 [56]; for TDBC, and Bradley et al. for TDBC [57]. The prolate Ag nanorods were designed such that their longitudinal LSPR, when immersed in a background relative permittivity similar to $\varepsilon_{\infty }$ of respective materials (i.e., 2.44, 2.74, and 3.13 for MEH-PPV, P3HT, and PTB7 respectively), was resonant with the center peak wavelength for the respective shell. To achieve this, we used short axes of 25 nm and long axes of 28 nm, 33 nm, and 37 nm for MEH-PPV, P3HT, and PTB7, respectively.

The scattering spectra were acquired by summing the transmission through a box of six frequency-domain power monitors surrounding the core-shell NP, located outside of the TFSF source. A three-dimensional frequency-domain power monitor was placed surrounding the core-shell NP, located inside of the TFSF source to monitor the total electric field within the hybrid NP. A three-dimensional refractive index monitor was overlaid on the total-field frequency-domain power monitor. For the total-field monitor, the fields were sampled at every other mesh point. Absorption in each layer was calculated using the following equation [43]: (1)$$A\left(x,y,z,\lambda \right)=n\left(x,y,z,\lambda \right)\alpha \left(x,y,z,\lambda \right){\left|\frac{E\left(x,y,z,\lambda \right)}{E_{0}}\right|}^{2}$$
where *n* is the real part of the complex index of refraction, *α* is the attenuation coefficient (4*πk*/*λ*), and |*E*/*E*_0_|^2^ is the normalized electric field intensity squared. The absorption was integrated spatially and normalized to the area illuminated (*S*_illum_; i.e., the cross-sectional area of the TFSF source in the plane normal to the direction of propagation) to obtain the spectral fraction of light absorbed in each layer:(2)$$A\left(\lambda \right)=\frac{\int_{x}\int_{y}\int_{z}A\left(x,y,z,\lambda \right)\partial x\partial y\partial z}{S_{\mathrm{illum}}}$$

The absorption in the core and shell were separated from each by using the refractive index monitor as a reference. Shell absorption enhancement values were obtained by simulating an identical core-shell nanostructure to the Ag core-conjugated polymer shell structure of interest, where the core was replaced with a dielectric material having a fixed relative permittivity of 1.77. The absorption in the shell with the Ag core was then divided by the absorption in the shell with the dielectric core to obtain the shell absorption enhancement factors.

Analytical Mie theory was additionally used to calculate the scattering cross-sections from bare and coated Ag spheres, as described by Bohren and Huffman [58]. The specific equations used for the calculations are provided in the online Appendix A. Comparison between Rayleigh, Mie, and FDTD methods is provided in Appendix A.

## 3. Results and Discussion

### 3.1. Modelling the Relative Permittivity of Conjugated Polymers

Figure 1b shows intensity-normalized absorption coefficients for three common conjugated polymers poly[2-methoxy-5-(2-ethylhexyloxy)-1,4-phenylenevinylene] (MEH-PPV); poly(3-hexylthiophene-2,5-diyl) (P3HT); poly([4,8-bis[(2-ethylhexyl)oxy] benzo[1,2-*b*:4,5-*b*’]dithiophene-2,6-diyl][3-fluoro-2-[(2-ethylhexyl)carbonyl]thieno[3,4-*b*] thiophenediyl]) (PTB7); and for a typical cyanine-based dye *J*-aggregate, 1,1′-diethyl-3,3′-di(4-sulfobutyl)-5,5′,6,6′-tetrachlorobenzimidazolo-carbocyanine (TDBC-JA). The linewidth of the *J*-aggregate absorption coefficient is 26 nm (0.09 eV), which is 4.5 to 6.7 times narrower than those of the three conjugated polymers, which range from 117 nm to 175 nm (0.44 eV to 0.55 eV), and predominantly features a single, narrow resonance peak. The three conjugated polymers each display broad absorption features, with multiple resonances arising from coupling of the electronic to the vibrational modes.

The frequency-dependent complex relative permittivity, ε(ω), of excitonic materials, such as *J*-aggregates of organic dyes, can be modelled by a single Lorentzian oscillator given by:(3)$$\varepsilon \left(\omega \right)=\varepsilon_{\infty }+\frac{f\omega_{0}^{2}}{\omega_{0}^{2}-\omega^{2}-i\gamma \omega }$$
where *ε*_∞_ is the high-frequency relative permittivity, *f* is the oscillator strength, *ω*_0_ is the resonance frequency of the oscillator, *γ* is the resonance linewidth, and *ω* is the frequency of electromagnetic radiation. Note that throughout the text, the resonance wavelength, *λ*_0_, will be reported instead of the resonance frequency of the oscillator, where: $\lambda_{0}=2\pi c/\omega_{0}$ (*c* is the speed of light), and the resonance linewidth will be represented in energy units (i.e., *ħγ*, in eV). For typical *J*-aggregates, *f* is in the range of 0.01 to 1.0 [12,13,59,60,61], and *ε*_∞_ is taken as either 1 (air) [30,31] or 1.77 (water) [28,29,59,60]. Unlike *J*-aggregates, conjugated polymers typically have optical line shapes with multiple resonances, have broader linewidths (0.05 eV to over 1 eV), large oscillator strengths (~0.02 to 5) for each individual oscillator, and larger *ε*_∞_ values (~2 to 4) [32,33,34,35]. Absorption transitions in conjugated polymers are designated by both their electronic energy level and vibrational energy level, *v*, where *v* is a positive integer (0,1,2,…). For steady-state, visible-light absorption, the conjugated polymer absorption spectra represent electronic energy level transitions from the singlet exciton ground state (*S*_0_) to the first singlet exciton excited state (*S*_1_) (Figure 1c). Absorption typically takes place from the first vibrational energy level in the *S*_0_ state (i.e., *v* = 0) to the *v*th vibrational energy level in the *S*_1_ state. As such, the vibrationally-dressed absorption peaks are usually classified as *S*_0_ → *S*_1_ 0-*v* (Figure 1c), with *v* increasing from the lowest to the highest energy vibrational energy level [62]. Thus, the relative permittivity of a conjugated polymer with 5 vibrational energy levels can be modeled by a summation of *m* = 5 Lorentzian oscillators [33,34,35,39], as given by:(4)$$\varepsilon \left(\omega \right)=\varepsilon_{\infty }+\sum_{v=0}^{m-1}\frac{f{\left[\omega_{0}+v\Delta \omega \right]}^{2}}{{\left[\omega_{0}+v\Delta \omega \right]}^{2}-\omega^{2}-i\gamma \omega }$$
where *m* is the number of oscillators, corresponding each discrete vibrational energy level, and Δ*ω* is the frequency spacing between oscillators. However, a simple summation of Lorentzian oscillators cannot fully describe the optical response of conjugated polymers. The relative strength of each 0-*v* transition is determined by the Franck-Condon principle, which states that the most probable transitions to occur are those with maximal orbital overlap between the initial (*S*_0_) and final (*S*_1_) states [38,62]. Thus, the relative permittivity in Equation (4) is modified by the Franck-Condon factor, *C_v_*, to give an accurate model of the optical response of conjugated polymers:(5)$$\varepsilon \left(\omega \right)=\varepsilon_{\infty }+\sum_{v=0}^{m-1}\frac{C_{v}f{\left[\omega_{0}+v\Delta \omega \right]}^{2}}{{\left[\omega_{0}+v\Delta \omega \right]}^{2}-\omega^{2}-i\gamma \omega }$$
where *C_v_*, is given by the intensity ratio of the 0-*v* transition to the 0-0 transition (i.e., $C_{v}=I_{0\to v}/I_{0\to 0}$), where $I_{0\to v}$ is given by:(6)$$I_{0\to v}={\left(\hbar \omega \right)}^{3}n_{f}^{3}\frac{S^{v}\exp\left(-S\right)}{v!}$$
where *n_f_* is the real part of the refractive index ($n_{f}=\sqrt{\varepsilon_{\infty }}$) and *S* is the Huang-Rhys factor [38,63]. For most conjugated polymers, the *S* parameter can be quite high, as large as 1. This is a valid model for excitonic materials exhibiting strong vibrational–electronic (i.e., vibronic) coupling.

The real and imaginary parts of the relative permittivity from the Lorentzian oscillator model of *J*-aggregates (Equation (3)) and from the Franck-Codon model of conjugated polymers (Equations (5) and (6)), are shown in Figure 1d,e for oscillator parameters selected based on those of a typical *J*-aggregate and conjugated polymer, respectively. The key Lorentzian oscillator parameters that are significantly different for conjugated polymers compared to *J*-aggregates are the oscillator strength (*f*), resonance linewidth (*ħγ*), number of oscillators (*m*), and high-frequency relative permittivity (*ε*_∞_). In this study, we systematically investigated the dependence of plasmon–exciton coupling strength on the thickness (*h*) and material parameters of the conjugated polymer shell (i.e., *f*, *ħγ*, *ε*_∞_, and *m*) in core-shell hybrid nanostructures.

### 3.2. Identifying Coupling Regimes

We began our study by employing optical-frequency material parameters similar to those of a typical *J*-aggregate (i.e., *f* = 0.05; *ħγ* = 0.10 eV; single oscillator (i.e., *m* = 1)) [28,29,31]; then we systematically varied the parameters to approach those of a typical conjugated polymer. Initially, we investigated core-shell nanostructures with a Ag core radius, *r*, of 25 nm and a shell thickness, *h*, of 5 nm, which was chosen as an upper limit for the thickness of typical *J*-aggregates adsorbed to Au or Ag nanostructures [12,13,30,31]. The resonance wavelength (*λ*_0_) was selected to overlap with the dipolar LSPR wavelength for the bare AgNP (i.e., *λ*_0_ = 420 nm), as identified by the scattering spectrum (Figure 2a). When coated by a shell with relative permittivity described by a single Lorentzian oscillator, the core-shell NP displayed splitting in its scattering spectrum, represented by two peaks which occurred at wavelengths red- and blue-shifted from the dipolar LSPR wavelength, with the red-shifted peak exhibiting greater intensity than the blue-shifted peak. This type of splitting is characteristic of plasmon–exciton coupling, and the two peaks are referred to as the plasmon–exciton hybrid modes [28,64].

To classify the strength of coupling between excitons within conjugated polymers and surface plasmons, we identified the coupling regime based on previous studies on plasmon–exciton and photon–exciton coupling [28,29,65]. There exist three dominant regimes for coupling strength between excitons and plasmons: the weak-coupling regime (which has also been called enhanced absorption [28], energy transfer [29], or the asymmetric-Fano regime [65,66]), the intermediate-coupling regime (i.e., induced-transparency regime [28], or anti-resonance Fano regime [65]), and the strong-coupling regime (i.e., plasmonic splitting [29], which is the plasmonic analogue of Rabi splitting) [28,29,65]. The coupling regime can be identified by comparing *ħ*Ω (the splitting energy) to the widths of the plasmon and exciton resonances (i.e., *γ*_sp_ and *γ*_ex_, respectively) [28,29,65]:(7)$$\hbar \Omega <\sqrt{\gamma_{\mathrm{sp}}\gamma_{\mathrm{ex}}}\;\;\;\;\mathrm{Weak}\;\mathrm{coupling}$$
(8)$$\sqrt{\gamma_{\mathrm{sp}}\gamma_{\mathrm{ex}}}<\hbar \Omega <\left(\gamma_{\mathrm{sp}}\vee \gamma_{\mathrm{ex}}\right)\;\;\;\;\mathrm{Intermediate}\;\mathrm{coupling}$$
(9)$$\hbar \Omega >\left(\gamma_{\mathrm{sp}}\wedge \gamma_{\mathrm{ex}}\right)\;\;\;\;\mathrm{Strong}\;\mathrm{coupling}$$

The splitting energy obtained from the scattering spectrum of the coated AgNP in Figure 2a (by taking the difference in energy between the blue-shifted peak and the red-shifted peak) was 290 meV, which, since it was less than *γ*_sp_ (378 meV) but greater than (*γ*_sp_
*γ*_ex_)^1/2^ (194 meV), confirmed that the core-shell nanostructure was within the intermediate coupling regime. Since the total absorption (Figure 2b) and scattering both displayed dips at the resonance wavelength, a transparency was induced within the hybrid core-shell nanostructure, which is a signature of the intermediate coupling regime [28]. The absorption in the core and shell was separated (Figure 2b), and the AgNP core absorption showed the same spectral features as the scattering from the core-shell structure. The absorption in the shell mostly resembled the Lorentzian lineshape of the uncoupled absorption transition, but was broadened and significantly enhanced (up to a factor of 160) relative to the absorption in an identical shell without the AgNP core. The shell absorption enhancement spectrum (i.e., absorption in the shell with the AgNP relative to absorption in the shell without the AgNP core; Figure 2c), which is a measure of the usefulness of employing plasmonic nanostructures for improving absorption within a dye or conjugated polymer shell, also showed the same spectral features as the scattering and core absorption. Thus, either the scattering spectrum, the core absorption spectrum, or shell absorption enhancement spectrum can be used to identify the plasmon–exciton hybrid modes. This indicates that the plasmon–exciton hybrid modes gave rise to the most useful absorption enhancements.

### 3.3. Single Lorentzian Oscillator Excitons

We initially studied very thin absorber shells, which have been the primary focus of core-shell plasmon–exciton coupling studies to date. This is because for core-shell nanostructures, typically only a monolayer of the *J*-aggregates adsorbs to the surface of the metallic nanoparticle. In addition, the photoluminescence of thin-films of *J*-aggregates is known to be drastically quenched relative to *J*-aggregates in the monolayer [67]. However, in many instances of metal nanoparticle-conjugated polymer heterostructures, the conjugated polymer layer is significantly thicker [49,50,51,52], partially because conjugated polymers undergo minimal photoluminescence quenching in the solid state. Therefore, we investigated the dependence of the shell thickness on the scattering and absorption from the core-shell NPs, initially using the single Lorentzian oscillator model for the relative permittivity of the shell material with *f* = 0.05 and *ħγ* = 0.10 eV (Figure 3). As *h* increased, the splitting energy between the hybridized modes increased (Figure 3a,d). As *h* increased to 20 nm and beyond, the coupling regime transitioned from induced transparency to strong coupling (Figure 3d), and a third peak started to develop at the resonance wavelength (Figure 3a). The formation of a three-peaked spectrum has been reported previously in studies of plasmon–exciton coupling, and the central peak was attributed to a shell mode that screens the metal core from the incident electromagnetic fields (Figure 3c, inset) [28]. As *h* was increased further, the shell mode became more dominant, suggesting that the Ag core became screened to a larger extent as the shell thickness increased. The splitting values began to saturate for *h* ≥ 30 nm (Figure 3d), because the localized electric field enhancement from the LSPR of the AgNP decayed to a value of unity at a distance of 40 nm from the NP surface (Figure 3c). Therefore, for shell thicknesses greater than 30 nm, only the portion of the shell within the first 30 nm was strongly coupled to the LSPR from the Ag core, and the remaining portion of the shell was weakly coupled or uncoupled and acted simply as an optical filter.

Although larger splitting energies were achieved for largest *h*, the smallest values of *h* resulted in the largest shell absorption enhancements, with maximal absorption enhancement factor greater than 200 for 2-nm-thick shells (Figure 3b). The absorption enhancement values decreased exponentially for increasing shell thickness, with 100-nm-thick shells experiencing enhancement factors near unity (i.e., absorption averaged over the entire shell area was almost the same with or without the Ag core) throughout the visible spectrum. The absorption enhancement values in the thinner shells were very large compared to many values reported in the plasmonic-enhanced photovoltaic literature [68], which we attributed to the thin shells absorbing only a small fraction of the incident light on their own (i.e., they were “optically-thin”), giving them the potential to become more absorptive when the local electric field intensity was enhanced. The plasmonic AgNPs increased the electric field intensity by factors of over 80 at their surfaces (Figure 3c, inset). Thus, the thinnest shells, which absorb weakest on their own, exhibited the largest absorption enhancements due to having most of their volume located closest to the most intense electric fields (Figure 3c). The shell absorption enhancement spectra resembled the scattering spectra, except that the absorption enhancement plots did not display the third shell mode at the resonance wavelength. Only the short and long wavelength plasmon–exciton hybrid modes contributed to the absorption enhancement in the shell.

We selected *h* = 40 nm for the remaining investigations, which is a modest thickness for conjugated polymer thin-films used for optoelectronic applications and is just beyond the LSPR decay length of the 25-nm-radius AgNP cores (Figure 3c). To define a more realistic conjugated polymer shell, we increased the exciton linewidth (*ħγ*) to values typical for conjugated polymers. There was only a weak dependence of the splitting energy on *ħγ* for both thin and thick shells (Appendix A). For thick shells, the splitting energy was large enough such that the core-shell structure remained within the strong coupling regime, up to *ħγ* = 0.38 eV, which is beyond the typical linewidth for conjugated polymers. Thus, the broad linewidth of the conjugated polymer shells had little influence on the strength of the coupling.

We next chose *ħγ* = 0.30 eV as representative of typical conjugated polymers [32,34,69], and varied *f* for *h* = 40 nm (Figure 4). Conjugated polymers typically have *f* ranging from ~0.02 to up to 5 for each oscillator [33,34,35]. As *f* increased, the splitting in the scattering and shell absorption enhancement spectra increased (Figure 4a,b). For *f* > 0.05, three peaks were observed in the scattering spectra, with the third, central, shell mode increasing in intensity with increasing *f* and becoming the dominant mode for *f* > 0.3. The normalized scattering of the short wavelength plasmon–exciton hybrid mode decreased in intensity with increasing *f*, completely disappearing by *f* > 0.5 (Figure 4a,c). However, the long wavelength mode continued to red-shift and remained relatively intense, even for large *f* values. The dominance of the long wavelength plasmon–exciton mode was further observed in the shell absorption enhancement spectra, in which the ratio of the long- to short wavelength mode continued to increase with increasing *f*, until the short wavelength mode no longer had any influence on the shell absorption enhancement for *f* ~ 1. The reason why the long wavelength plasmon–exciton hybrid mode had greater scattering intensity and, thus, shell absorption enhancement, was because scattering efficiency increases with increasing relative permittivity (see Appendix A), and for *λ* > *λ*_0_, the relative permittivity of both the shell and the AgNP core increase. This gave rise to increased scattering intensity for *λ* > *λ*_0_, and decreased scattering intensity for *λ* < *λ*_0_. Because the scattering intensity determines the shell absorption enhancement, the long wavelength plasmon–exciton hybrid mode was more important for achieving significant absorption enhancement within the conjugated polymer shell.

Since the short wavelength mode vanished for large *f*, and overlapped significantly with the third shell mode, the splitting energy here was calculated as twice the difference between the energy of the long wavelength mode from the resonance energy. Under these conditions, the core-shell structure was in the strongly coupled regime for all *f* > 0.05. Splitting was also observed in the shell absorption spectra for *f* > 0.05, which is typically only observed within the strong coupling regime (see Appendix A) [28]. We observed maximal splitting values of 2100 meV for *f* = 2.5, which is among the largest predicted to date for plasmon–exciton coupling [26,61,70]. For more modest *f* in the range of 0.3 to 0.6, the splitting energy was still in the high range of 900 meV to 1200 meV. As such, Ag-conjugated polymer core-shell nanostructures are excellent candidates for achieving strongly coupled plasmon–exciton systems.

Since the short wavelength mode vanished for large *f*, and overlapped significantly with the third shell mode, the splitting energy here was calculated as twice the difference between the energy of the long wavelength mode from the resonance energy. Under these conditions, the core-shell structure was in the strongly coupled regime for all *f* > 0.05. Splitting was also observed in the shell absorption spectra for *f* > 0.05, which is typically only observed within the strong coupling regime (see Appendix A) [28]. We observed maximal splitting values of 2100 meV for *f* = 2.5, which is among the largest predicted to date for plasmon–exciton coupling [26,61,70]. For more modest *f* in the range of 0.3 to 0.6, the splitting energy was still in the high range of 900 meV to 1200 meV. As such, Ag-conjugated polymer core-shell nanostructures are excellent candidates for achieving strongly coupled plasmon–exciton systems.

So far, we have assumed a fixed value of 1.77 for *ε*_∞_ for the conjugated polymer shell. Previous plasmon–exciton studies involving *J*-aggregates as excitonic materials have typically investigated core-shell nanostructures in solution [28,29,59,60] or single nanostructures in air [30,31] such that *ε*_∞_ has usually been fixed at values of either 1.77 (water) or 1 (air). However, realistic conjugated polymers have larger *ε*_∞_ values (typically ranging between 2 and 4) [32,33,34,35]. Thus, we varied *ε*_∞_ for core-shell structures with single Lorentzian oscillator shells, having *f* = 0.10, *ħg* = 0.30 eV, and *h* = 40 nm (Figure 5). By increasing *ε*_∞_, the LSPR of the AgNP core red-shifted away from the resonance of the excitonic transition (see Appendix A). To ensure that the plasmon and exciton were resonant, we first determined the red-shifted LSPR wavelengths for bare AgNPs immersed in background relative permittivities, *ε*_bkgd_, equal to each *ε*_∞_. The *λ*_0_ of the conjugated polymer shells were then adjusted such that the excitons were resonant with the LSPRs for each *ε*_∞_ (listed in Figure 5a). However, the process of shifting the resonance wavelength with increasing *ε*_∞_ also resulted in a reduction in the amplitude of the relative permittivity, which effectively changed the transition dipole moment. In order to investigate only the influence of *ε*_∞_ on the coupling strength, we thus used a scaling factor for *f* (i.e., *f’*) in Equation (3) such that the amplitude of the imaginary component of the relative permittivity, *ε*_imag_, was constant as *ε*_∞_ and *λ*_0_ increased. The normalized scattering is shown as a function of detuning, where detuning is the energy difference away from *λ*_0_. As *ε*_∞_ increased, the splitting energy decreased due to the increased screening of the electric fields in the shells with higher permittivities. Therefore, compared to *J*-aggregates, the *ε*_∞_ of conjugated polymers decreased the coupling strength. Despite the decrease in splitting energy for larger *ε*_∞_, strong coupling was still achievable for *ε*_∞_ up to 2.5, and the intermediate coupling regime was achievable for *ε*_∞_ of 2.5 to 3. The decrease in splitting energy for increasing *ε*_∞_ is also a factor for the significantly lower splitting energies observed in inorganic plasmon–exciton coupling studies [4,8].

### 3.4. Vibrationally-Dressed Excitonic States

To use a more realistic model of the optical properties of conjugated polymers, we used the Franck-Condon vibronic progression, using Equations (5) and (6) (Figure 6). The number of vibronic modes represented as individual oscillators for conjugated polymers typically vary from 1 to 5 [32,33,34,35,55,71], and it is common for *ħγ* to be fixed or similar for all of the oscillators. Typically, as the vibronic modes become more well-resolved, *ħγ* for each individual oscillator becomes reduced relative to conjugated polymers with only one, broad absorption feature (single oscillator) [69]. Therefore, we fixed *ħγ* = 0.15 eV, *f* = 0.10, and *ε*_∞_ = 3.0 for all oscillators and investigated scattering from core-shell nanostructures with *h* = 40 nm (Figure 6). We used a Huang-Rhys factor (*S*) of 1, which is typical for common conjugated polymers [38], and fixed the dominant transition (i.e., the 0-1 mode; see Figure 1e) of the conjugated polymer shell to be resonant with the uncoupled LSPR of the AgNP with *ε*_bkgd_ = 3.0 (i.e., 500 nm). For a single oscillator, we observed a large splitting of 423 meV consistent with the *ħγ*, *ε*_∞_, *f*, and *h* values used in this case. As the number of oscillators increased, the splitting between the shortest and longest wavelength modes increased, and additional shell modes were observed at wavelengths between the hybrid plasmon–exciton modes. For *m* oscillators, *m*+2 peaks were observed, corresponding to the two dominant hybrid modes (shortest and longest wavelength peaks) and *m* center shell modes. The *m* shell modes were all comparable in intensity, while the longest wavelength hybrid mode was the most intense, and the shortest wavelength hybrid mode decreased in intensity with increasing *m*. The maximum splitting energy calculated was 820 meV for 5 oscillators (Figure 6b), which was 33% of the resonance energy (2.48 eV), placing this coupling strength into the ultrastrong regime [72,73,74]. These results demonstrate that the presence of vibronic modes in conjugated polymers can lead to large splitting energies and, therefore, (ultra)strong coupling between excitons and plasmons. We note that by using a summation of Lorentzian oscillators instead of the Franck-Condon progression to model the relative permittivity of conjugated polymers, the coupling strengths are overestimated by more than 20% (Appendix A).

Finally, we investigated the coupling between conjugated polymer shells with realistic dispersive relative permittivities obtained from experimental studies and LSPRs from Ag nanorods [75]. Prolate Ag nanorods were chosen due to the high degree of spectral tunability of the longitudinal LSPR by varying the length of the nanorod, allowing a plasmonic mode to be resonant with the conjugated polymer absorption [76]. The nanorods were modeled such that their longitudinal LSPR was the only mode excited (by controlling excitation polarization), and was resonant with the central peak wavelength of the various conjugated polymers (see Methods and Figure 7). For these core-shell structures, *h* = 40 nm, and for the nanorod, the long axis length was varied to tune the LSPR wavelength, and the short axis had a fixed radius of 25 nm. For the three conjugated polymers studied here, MEH-PPV, P3HT, and PTB7, all of the scattering spectra for the core-shell structures showed intense, long-wavelength plasmon–exciton hybrid modes and weak, short-wavelength hybrid modes, with multiple broad shell modes occurring at wavelengths between the hybrid modes (Figure 7a–c). The scattering spectra were qualitatively very similar to those calculated from the model conjugated polymers with *m* = 5 oscillators (Figure 6a).

As with prior reports on plasmon-enhanced conjugated polymer structures [30,43,44,45,46,59,60,61,77], we observed the largest absorption enhancement in the conjugated polymer shell at wavelengths red-shifted from the absorption of the uncoupled polymers (Figure 7d–f). This red-edge absorption enhancement, which has been frequently reported in the literature, can now be attributed to the long wavelength hybrid mode of a coupled plasmon–exciton system. The splitting energies observed for these polymers were 736 meV, 692 meV, and 834 meV for MEH-PPV, P3HT, and PTB7, respectively, placing each of these core-shell structures well within the strong coupling regime. In fact, because the splitting energies were between 29% and 42% of the resonance energies (i.e., 2.53 eV, 2.23 eV, and 2.0 eV for MEH-PPV, P3HT, and PTB7, respectively) in the uncoupled conjugated polymers, the Ag nanorod-conjugated polymer core-shell structures were in the ultrastrong coupling regime [17,26,70]. We note that the intensities of the long wavelength scattering peaks were large considering the 40-nm-thick conjugated polymer shells (Appendix A), ranging from 23% to 64% of the scattering intensity from the bare Ag nanorods. We observed a similar intense long wavelength scattering peak in prior work for conjugated polymers coated onto plasmonic metasurfaces, and called the peak “Absorption-Induced Scattering” (AIS) resulting from coupling between excitons and scattering modes [43]. In that study, the coupling between modes was not as clear due to the multiple broad resonances associated with both the conjugated polymer coatings and the plasmonic metasurfaces. However, the splitting observed was as large as 1400 meV, much greater than the linewidths of the exciton and plasmon modes. In this study, we confirmed that AIS is in fact the long wavelength plasmon–exciton hybrid mode for conjugated polymer-metal nanostructures and is a signature of strong coupling. Thus, conjugated polymers serve as excellent candidates for applications requiring strong coupling between plasmons and excitons.

## 4. Conclusions

In conclusion, we have systematically investigated plasmon–exciton coupling in Ag-conjugated polymer core-shell nanostructures using numerical finite-difference time-domain and analytical Mie theory calculations. Compared to a thin (less than 5 nm) organic dye *J*-aggregate shell, which has been the focus of most prior theoretical and experimental studies, we have demonstrated that it is possible to achieve strong plasmon–exciton coupling with conjugated polymers, despite their larger linewidths and high-frequency relative permittivities. This is attributed to a combination of their large oscillator strengths, multiple vibronic resonances, and the practicality of employing physically thicker shells. We calculated splitting energy values ranging from 400 meV to greater than 1000 meV for conjugated polymer-coated AgNPs within the strong coupling regime. We found that pronounced scattering occurring at wavelengths longer than the absorption band edge wavelength of the conjugated polymer (previously identified as Absorption-Induced Scattering) is a signature of strong plasmon–exciton coupling. Strong coupling led to the highest absorption enhancement in the conjugated polymer shell at wavelengths red-shifted from the absorption band edge because the long wavelength hybrid mode was dominant for the strongest coupling. These findings will help stimulate interest in designing conjugated polymer optoelectronic devices enhanced by plasmonic nanostructures, including solar cells, light-emitting diodes, and spasers, and the conditions to achieve strongly coupled plasmons and excitons using these materials have been identified.

## Figures and Tables

**Figure 1 polymers-12-02141-f001:**
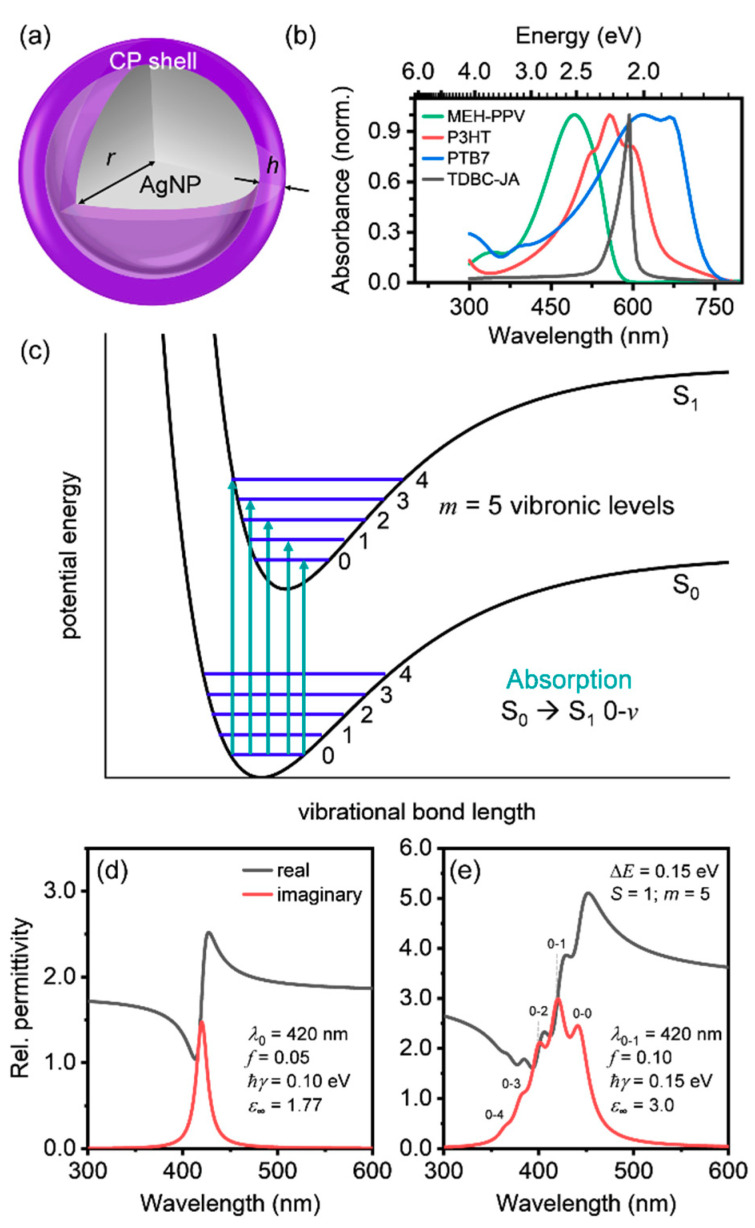
(**a**) Core-shell nanoparticle structure used to investigate coupling between excitons in conjugated polymers (CPs) and localized surface plasmon resonances (LSPRs). The silver nanoparticle (AgNP) core radius is given by *r*, and the CP shell thickness is given by *h*. (**b**) Normalized absorption coefficient of various conjugated polymers (MEH-PPV, P3HT, and PTB7) and a typical cyanine dye *J*-aggregate (TDBC-JA; see abbreviations section for full names of polymers and dye). (**c**) Schematic of potential energy curves of a simple anharmonic oscillator in the ground (*S*_0_) and excited (*S*_1_) states as functions of the bond length. The solid blue lines (0,1,2,...) represent the vibrational energy levels (*v*) associated with each electronic energy level. Absorption transitions are shown as vertical lines from *S*_0_ → *S*_1_ 0-*v*. (**d**) Real and imaginary parts of the relative permittivity for a single Lorentzian oscillator used as the basis for modeling *J*-aggregate optical properties. (**e**) Real and imaginary parts of the relative permittivity for a model conjugated polymer comprised of *m* = 5 oscillators. Symbols used: 0-*v* = absorption transition from *S*_0_ → *S*_1_, into the *v*th vibrational level; *λ*_0_ = resonance wavelength; *λ*_0-1_ = resonance wavelength for 0-1 transition; *f* = oscillator strength; *ħ**γ* = resonance linewidth; *ε*
_∞_ = high-frequency relative permittivity; Δ*E* = spacing between oscillators; *S* = Huang-Rhys factor.

**Figure 2 polymers-12-02141-f002:**
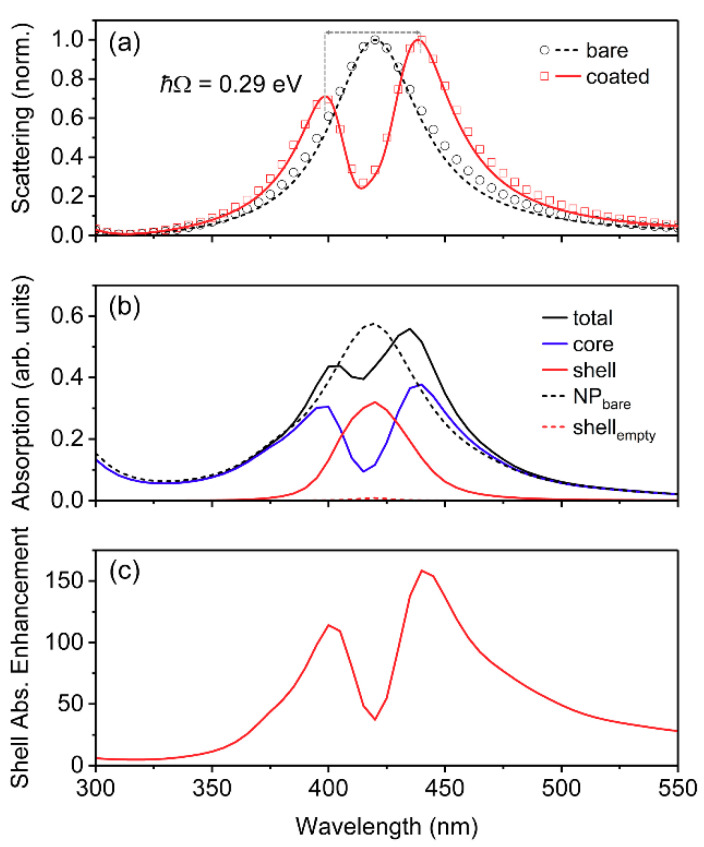
(**a**) Normalized scattering, (**b**) absorption, and (**c**) shell absorption enhancement from core-shell nanoparticles with shell relative permittivity described by a single Lorentzian oscillator. The AgNP radius and shell thickness were *r* = 25 nm and *h* = 5 nm, respectively, and the shell parameters used were: *λ*_0_ = 420 nm; *f* = 0.05; *ħγ* = 0.10 eV; *ε*_∞_ = 1.77. Note that *ħ*Ω represents the splitting energy between the plasmon–exciton hybrid modes, obtained from the scattering spectra, as shown in (**a**). In (**a**), solid lines represent normalized scattering cross-section calculated using analytical Mie theory, and open symbols represent normalized scattering cross-section calculated using numerical finite-difference time-domain (FDTD) simulations. In (**b**,**c**), only results from the FDTD simulations are shown, using Equation (2) to calculate the absorption in each layer.

**Figure 3 polymers-12-02141-f003:**
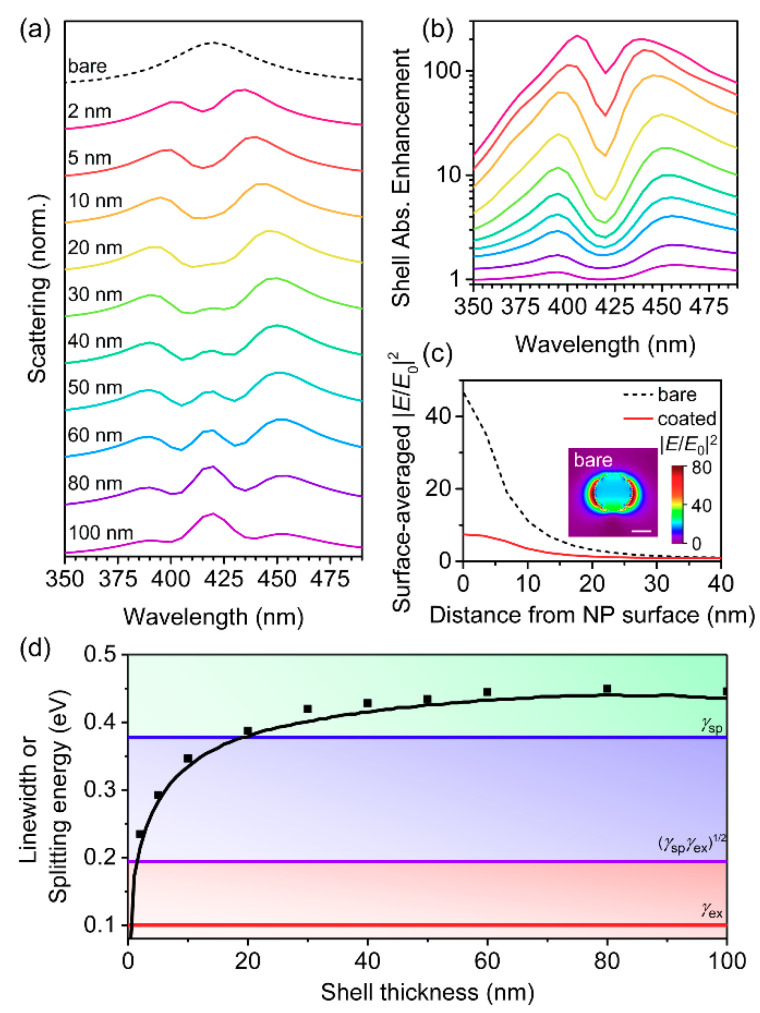
Variation of shell thickness, *h*, for core-shell structures with a single Lorentzian oscillator shell with *ε*_∞_ = 1.77, *λ*_0_ = 420 nm, *f* = 0.05, and *ħγ* = 0.10 eV. (**a**) Normalized scattering and (**b**) shell absorption enhancement calculated using FDTD simulations. The colors used in (**b**) are the same as those used in (**a**). (**c**) Surface-averaged electric field intensity enhancement, |*E*/*E*_0_|^2^, for a bare AgNP and a core-shell structure with *h* = 5 nm. Inset is the |*E*/*E*_0_|^2^ enhancement profile taken from a plane parallel to the excitation polarization at *λ* = 420 nm (scale bar is 25 nm). (**d**) Splitting energy for hybrid plasmon–exciton modes extracted from the scattering spectra using FDTD simulations (symbols) and analytical Mie theory (black line). The background shading represents the coupling regimes (green—strong coupling; blue—intermediate coupling; red—weak coupling) as defined by the linewidths of the surface plasmon (*γ*_sp_—blue line), the exciton (*γ*_ex_—red line), and the relationship: $\sqrt{\gamma_{\mathrm{sp}}\gamma_{\mathrm{ex}}}$ (purple line), given in Equations (7)–(9).

**Figure 4 polymers-12-02141-f004:**
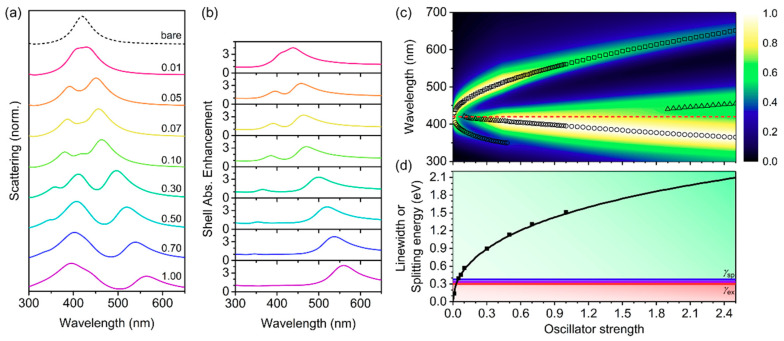
Variation of oscillator strength, *f*, for core-shell structures with a single Lorentzian oscillator shell with *h* = 40 nm, *ε*_∞_ = 1.77, *λ*_0_ = 420 nm, and *ħγ* = 0.30 eV. (**a**) Normalized scattering; and (**b**) shell absorption enhancement calculated using FDTD simulations. (**c**) Normalized scattering cross-section calculated using Mie theory. The colorbar represents the normalized scattering intensity, the open symbols represent the peaks of the dominant modes, and the red dashed line represents *λ*_0_. (**d**) Splitting energy for hybrid plasmon–exciton modes calculated using FDTD simulations (symbols) and Mie theory (black line). Splitting energy here was calculated as $\hbar \Omega =(E_{0}-E_{\mathrm{lower}})\times 2$, where *E*_0_ is the resonance energy (2.95 eV) and *E*_lower_ is the energy of the long wavelength plasmon–exciton hybrid mode. The background shading represents the coupling regimes (green—strong coupling; blue—intermediate coupling; red—weak coupling) as defined by the linewidths of the surface plasmon (*γ*_sp_—blue line), the exciton (*γ*_ex_—red line), and the relationship: $\sqrt{\gamma_{\mathrm{sp}}\gamma_{\mathrm{ex}}}$ (purple line), given in Equations (7)–(9).

**Figure 5 polymers-12-02141-f005:**
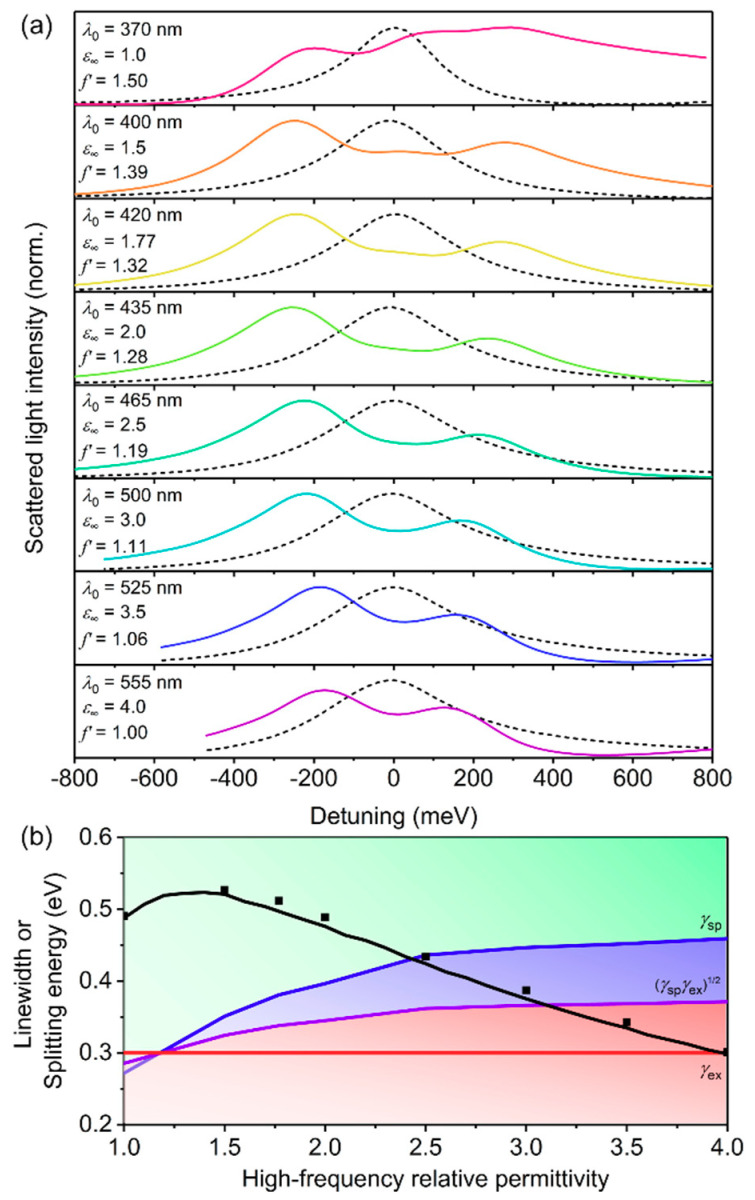
Variation of high-frequency relative permittivity, *ε*_∞_, for core-shell structures with a single Lorentzian oscillator shell with *h* = 40 nm, *f* = 0.10, and *ħγ* = 0.30 eV. (**a**) Normalized scattering spectra calculated using FDTD simulations. Note that *λ*_0_ was selected as the LSPR wavelength of bare AgNPs in a background relative permittivity equal to *ε*_∞_ (black dashed line). The *f’* values listed are the scaling factors by which *f* was divided in order to have normalized amplitudes of the imaginary components of the relative permittivity to account for the shifted *λ*_0_. (**b**) Splitting energy for hybrid plasmon–exciton modes calculated using FDTD simulations (symbols) and Mie theory (black line). The background shading represents the coupling regimes (green—strong coupling; blue—intermediate coupling; red—weak coupling) as defined by the linewidths of the surface plasmon (*γ*_sp_—blue line), the exciton (*γ*_ex_—red line), and the relationship: $\sqrt{\gamma_{\mathrm{sp}}\gamma_{\mathrm{ex}}}$ (purple line), given in Equations (7)–(9).

**Figure 6 polymers-12-02141-f006:**
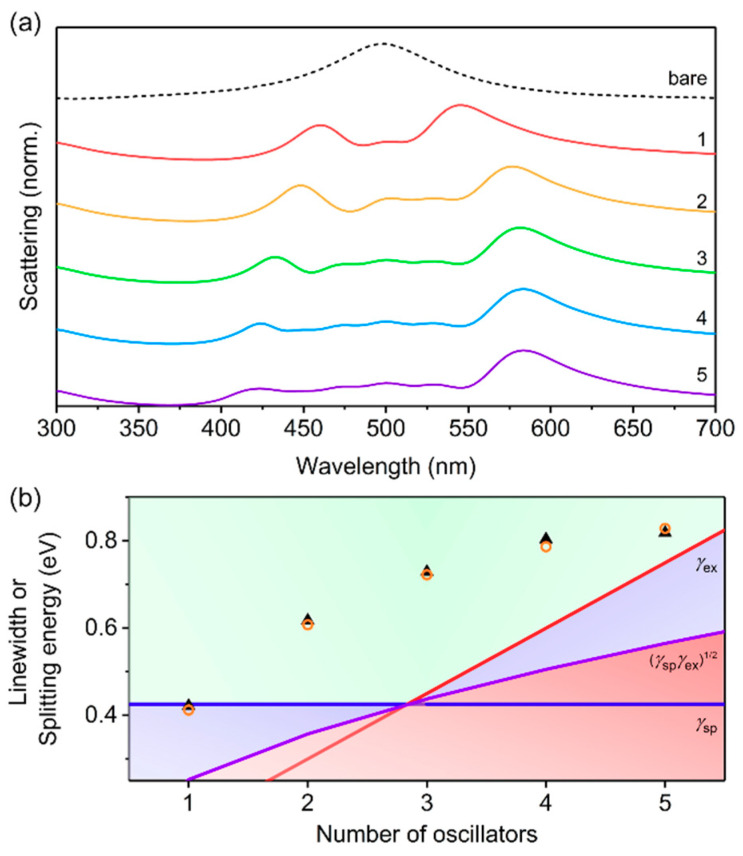
Variation of number of oscillators, *m*, for core-shell structures with shell relative permittivity described by the Franck-Condon progression (Equations (5) and (6)) and with *h* = 40 nm, *f* = 0.10; *ħγ* = 0.15 eV; Δ*E* = 0.15 eV; *ε*_∞_ = 3.0; and *S* = 1. The most intense oscillator (i.e., the 0-1 transition for *m* ≥ 2) was resonant with the uncoupled LSPR of the AgNP (i.e., 500 nm), with additional vibrational modes either blue- or red-shifted from the uncoupled LSPR. (**a**) Normalized scattering spectra calculated using FDTD simulations. (**b**) Splitting energy for hybrid plasmon–exciton modes calculated using FDTD simulations (black solid triangles) and Mie theory (orange open circles). The background shading represents the coupling regimes (green—strong coupling; blue—intermediate coupling; red—weak coupling) as defined by the linewidths of the surface plasmon (*γ*_sp_—blue line), the exciton (*γ*_ex_—red line), and the relationship: $\sqrt{\gamma_{\mathrm{sp}}\gamma_{\mathrm{ex}}}$ (purple line), given in Equations (7)–(9). The splitting energy was obtained from the scattering spectra as the difference in energy between the highest and lowest energy hybrid modes.

**Figure 7 polymers-12-02141-f007:**
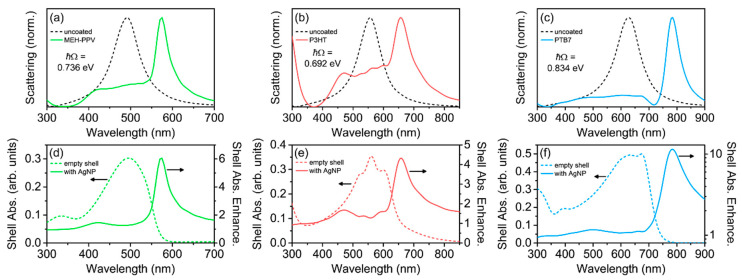
(**a**–**c**) Normalized scattering and (**d**–**f**) shell absorption (left axis) and absorption enhancement (right axis) spectra for prolate Ag nanorod core-shell structures with dispersive relative permittivities for common conjugated polymer materials used as the shells with *h* = 40 nm. The nanorod short axes were 25 nm and the long axes were: 28 nm (MEH-PPV); 33 nm (P3HT); and 37 nm (PTB7). Absorption and scattering calculations were conducted using FDTD simulations. Splitting energy here was calculated as $\hbar \Omega =(E_{0}-E_{\mathrm{lower}})\times 2$, where *E*_0_ is the resonance energy and *E*_lower_ is the energy of the long wavelength plasmon–exciton hybrid mode.

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
