# Peer review of "Strong Plasmon–Exciton Coupling in Ag Nanoparticle—Conjugated Polymer Core-Shell Hybrid Nanostructures"

_polymers, 2020, doi:10.3390/polym12092141_

Round 1

Reviewer 1 Report

This manuscript describes the calculated optical interaction (particularly coupling regime) between conjugated polymers with large linewidths and high-frequency relative permittivities and localized surface plasmon resonance of Ag nanospheres. These interactions can be attributed to the strong plasmon-exciton coupling and the authors suggest that they can be achieved by large oscillator strength and physically thicker shells of conjugated polymers. This study can support the author’s previous paper (Nat. Commun. 2015, 6, 7899.). And the calculation methods based on Mie theory, which are mainly described in Supporting Information, are appropriate. Thus, the reviewer suggests that this manuscript should be published in Polymers as is.  

Author Response

We thank the reviewer for taking the time to read through and review our manuscript, and for recommending publication as-is. We greatly appreciate the reviewer’s time and feedback.

Reviewer 2 Report

Authors have investigated plasmon-exciton coupling in silver conjugated polymer core-shell nanostructures using FDTD and Mie theory. The main contribution is that authors have showed  strong plasmon-exciton coupling with conjugated polymers even though thy have large spectral width (line width). This is a comprehensive study and further builds on some of the past work carried out it in this are such as Nat.Communication (ref 41). I recommend the manuscript to be published pending minor revisions.

1) Authors may need to provide more clarity on the line width limit (a value where the coupling reduced to half the value) rather than using the broad term 'large spectral width'. More quantitative information can be provided.

2) Page 2, line 88: Please mention if the simulation was done in 2D or 3D? I believe it is 3D based on Figure 1(a)

3) Figure 1B represents line width in terms of wavelength and in some places of the text it is mentioned in terms of eV. Please make it consistent.

4) Page 3, Line 94: Authors have used TFSF source dimensions 100 nm larger than the diameter of the NP. Please provide the reason. How about large source dimension? Would it affect the results.

5) Page 3, line 97: How did authors create a 1.99 fs pulse?

6) Page 4, line 144: Please provide FWHM values for the spectral width (line width) to understand how narrow it is compared to the conjugated polymers

7) Page 12, line 365: Please provide reference for Ag nanorod?

8) Page 2, line 54: The light confinement is achieved with microcavities and microstructures.  It is recommended to include more references about 'microcavities and microstructures' such as
Scientific Reports volume 3, Article number: 1577 (2013) , Applied Optics Vol. 49, Issue 11, pp. 2099-2104 (2010)
